# Effect of Calcination Temperature and Time on the Synthesis of Iron Oxide Nanoparticles: Green vs. Chemical Method

**DOI:** 10.3390/ma16051798

**Published:** 2023-02-22

**Authors:** Johar Amin Ahmed Abdullah, Mercedes Jiménez-Rosado, Antonio Guerrero, Alberto Romero

**Affiliations:** 1Departamento de Ingeniería Química, Escuela Politécnica Superior, Universidad de Sevilla, 41011 Sevilla, Spain; 2Departamento de Ingeniería Química, Facultad de Química, Universidad de Sevilla, 41012 Sevilla, Spain

**Keywords:** *Phoenix dactylifera L.*, nanoparticles, green synthesis, chemical synthesis, iron oxide, calcination temperature

## Abstract

Nowadays, antioxidants and antibacterial activity play an increasingly vital role in biosystems due to the biochemical and biological reactions that involve free radicals and pathogen growth, which occur in many systems. For this purpose, continuous efforts are being made to minimize these reactions, including the use of nanomaterials as antioxidants and bactericidal agents. Despite such advances, iron oxide nanoparticles still lack knowledge regarding their antioxidant and bactericidal capacities. This includes the investigation of biochemical reactions and their effects on nanoparticle functionality. In green synthesis, active phytochemicals give nanoparticles their maximum functional capacity and should not be destroyed during synthesis. Therefore, research is required to establish a correlation between the synthesis process and the nanoparticle properties. In this sense, the main objective of this work was to evaluate the most influential process stage: calcination. Thus, different calcination temperatures (200, 300, and 500 °C) and times (2, 4, and 5 h) were studied in the synthesis of iron oxide nanoparticles using either *Phoenix dactylifera L. (PDL)* extract (green method) or sodium hydroxide (chemical method) as the reducing agent. The results show that calcination temperatures and times had a significant influence on the degradation of the active substance (polyphenols) and the final structure of iron oxide nanoparticles. It was found that, at low calcination temperatures and times, the nanoparticles exhibited small sizes, fewer polycrystalline structures, and better antioxidant activities. In conclusion, this work highlights the importance of green synthesis of iron oxide nanoparticles due to their excellent antioxidant and antimicrobial activities.

## 1. Introduction

The eco-friendly green synthesis of iron oxide has increased in popularity in recent years due to its effectiveness, low cost, non-toxicity, and environmentally friendly nature. Furthermore, these methods can produce nanoparticles with high crystallinity in a variety of sizes, morphologies, and high magnetic properties [1,2,3,4,5,6,7,8]. These approaches are based on an active substance (phenolic compounds) present in plants, which acts as a reducing agent on metal salts to obtain metal oxide nanoparticles [9,10,11]. This led to the improved production of nanoparticles with unique, specific, and magnetic properties [12,13,14,15]. Nevertheless, nanoparticles based on phenolic compounds need further investigation to evaluate factors such as extraction methods, reactant concentrations, pH solution, metal salt, and reaction and calcination temperature and time [16,17]. In this way, the control of nanoparticle size and functionality is still a challenge in green methods.

During synthesis, the mixing of reducing agents with precursors results in rapid nucleation of the nanoparticles, followed by their aggregation. Therefore, it is evident that the rapid nucleation at the nanoparticle formation results in larger nanoparticle aggregates, heterogeneous crystal phases, and lower efficiency. This effect could be resolved by applying a post-processing technique, such as thermal treatment [18,19]. The thermal treatment can eliminate impurities, control the chemical phase of the capping agent, and determine the final phase, purity, and functionality of the final product. In this sense, homogenous or heterogeneous crystal phases could be modified by the final calcination treatment [18,19,20]. Nevertheless, this treatment has different effects on green and chemical synthesis. Thus, it can degrade the functional groups provided by green synthesis, reducing nanoparticle functionality [20]. Conversely, the chemical methods require a high calcination temperature to remove any excessive or secondary agents formed during the synthesis [18]. Thus, it is necessary to specify the calcination temperature and time to achieve the desired properties.

Natural and synthetic iron oxide nanoparticles are available in various forms and quantities, including iron oxides, hydroxides, and/or oxide-hydroxides complexes. Their atomic composition contains the same elements (Fe, O, and OH), although their crystal structures and iron valence may be different. However, the most important iron oxide nanoparticles (Fe_x_O_y_-NPs) are based on magnetite (Fe_3_O_4_-NPs), hematite (Fe_2_O_3_-NPs), and maghemite (γ-Fe_2_O_3_-NPs) [21,22,23,24,25,26,27,28,29], since they have a significant impact on the physicochemical and functional properties [30]. Iron oxide nanoparticles range in diameter from 1–100 nm and display a high surface-area-to-volume ratio, providing them with a significantly enhanced binding capacity and high dispersibility in solutions. Moreover, iron oxide nanoparticles < 20 nm in size display superparamagnetic properties [31]. These properties, in addition to their biocompatibility and nontoxicity, favor their potential application in biomedicine, environmental remediation, defense and aerospace, electronics, construction, healthcare, automotive textiles, agriculture, and the food industry [32,33].

Smaller iron oxide nanoparticles exhibit excellent antioxidant and antibacterial activities. Thus, they have a high inhibitory ability against foodborne pathogen growth, including *Escherichia coli* and *Staphylococcus aureus* [34]. This capacity to kill bacteria is due to the production of reactive oxygen species (^•^OH, ^•^O_2_^−^), which damage DNA and the bacteria protein, causing mitochondrial dysfunction [31,35,36]. Magnetic iron oxide nanoparticles have also been used to isolate and identify the DNA of pathogens in dairy products [37].

In biosystems, antioxidants and antibacterial activity play a crucial role. The biochemical and biological reactions inside living cells generate lethal free radicals, which are found in a variety of biological systems. However, little information is available regarding the study of magnetic iron oxide nanoparticles as antioxidants and antibacterial agents. Therefore, the effects of iron oxide nanoparticles on biochemical reactions should be investigated, as it is necessary to understand their antioxidant and antibacterial activity. They have been reported in previous works, concluding that several factors (e.g., particle size and magnetic behavior) could alter these properties [38]. Nevertheless, no study has extensively examined the effect of photochemical degradation on the efficiency and functionality of nanoparticles, especially those prepared from *Phoenix dactylifera L.* extract. The active photochemical responsible for donating the nanoparticle to its maximum functional capacity should not be degraded or destroyed during the synthesis. Therefore, the main objective of this work was to provide a novel comparison between the green and chemical synthesis of iron oxide nanoparticles and to demonstrate the impact of the calcination stage on their properties and functionality. In this way, *Phoenix dactylifera L. (PDL)* was used for its polyphenol content, as reported in previous works [39,40]. Factors like calcination temperatures and times, which may influence iron oxide nanoparticles’ properties, were investigated. Thus, iron oxide nanoparticles were synthesized by colloidal precipitation using two reducing agents: polyphenols extracted from *PDL* (green method, GS-NPs) and sodium hydroxide (chemical method, CS-NPs). Nanoparticles of both approaches were investigated under different calcination temperatures (200, 300, 500 °C) and times (2, 3, 5 h) and were further characterized by XRD, FTIR, SEM, and TEM. Furthermore, the obtained iron oxide nanoparticles were tested for their antioxidant activity towards DPPH and their antibacterial activity towards *Staphylococcus aureus* (*S. aureus*) and *Escherichia coli* (*E. coli*).

## 2. Materials and Methodology

### 2.1. Materials

In this study, the following chemicals: iron chloride hexahydrate (FeCl_3_·6H_2_O 98%), sodium hydroxide (NaOH), ethanol (CH_3_CH_2_OH 99.9%), methanol (CH_3_OH 99.9%), gallic acid (C_7_H_6_O_5_), 2,2-diphenyl-1-picrylhydrazyl (DPPH), and dimethyl sulfoxide anhydrous (DMSO) (C_2_H_6_SO 99.9%) were all supplied by Sigma Aldrich (Darmstadt, Germany) and used. The experiments were performed employing only analytical-grade chemicals and reagents.

Polyphenols were extracted from *PDL*, which was collected from Seville (Spain), following a previous protocol [41,42]. Briefly, leaves were dried for 44 days at 22 ± 3 °C and 35% RH. Polyphenols were extracted by the Soxhlet method with distilled water (water: leaf powder ratio 10:1) for 8 h. The extract had a total polyphenol content (TPC) of 39 ± 2 mg GAE/g extract.

### 2.2. Nanoparticle Synthesis

Iron oxide nanoparticles were produced in two ways: greenly (GS-NPs) by adding 20 mL of *PDL* extract dropwise into 20 mL of iron chloride (FeCl_3_·6H_2_O) 1 M solution, and chemically (CS-NPs) by adding 20 mL of NaOH into 20 mL of FeCl_3_·6H_2_O 1 M solution. NaOH 5 M was added to adjust the pH of the reaction to 7.5. The resulting solutions were heated on a hot plate at 50 °C under stirring for 2 h. Thereafter, the mixture was filtrated using Whatman nº 1 filter paper, and then, a series of washings with distilled water was conducted to eliminate contaminants and foreign particles. The precipitates were subjected to pre-treatment at 100 °C for 8 h in an oven. Then, they were calcined. This last step was evaluated using different calcination temperatures (200, 300, and 500 °C) for different times (2, 4, and 5 h). A more detailed description of this synthesis is provided in previous works [41,42,43].

### 2.3. Nanoparticle Characterization

#### 2.3.1. X-ray Diffraction (XRD)

The XRD patterns were acquired using a Brand diffractometer (Bruker model D8 advance A25 diffractometer with Cu anode, Bruker) for the confirmation of the crystalline phase. The nanoparticle diffractograms were performed between 2θ° = 15 and 70°. The X-ray with a beam power of 40 kV and a current of 30 mA made contact with the samples at 0.015° (pitch angle) and 0.1 s (passing time), while the samples were rotating at 30 rpm [44]. As described in previous studies [41,42], the size and crystallinity of the magnetic iron oxide nanoparticles were calculated using the Debye-Scherrer formula.

#### 2.3.2. Fourier Transform Infrared Spectroscopy (FTIR)

A DTGS-KBr sensor was used in transmittance mode to perform FTIR spectroscopy on a Hyperion 100 Spectrometer (Bruker, Billerica, MA, USA). The iron oxide nanoparticles were characterized from 4000 to 400 cm^−1^ to gather their structural information. Fe-O bonds in Fe_x_O_y_ can be identified in the fingerprint region (800–400 cm^−1^) [41,42].

#### 2.3.3. Scanning Electron Microscopy (SEM)

An SEM study was performed for the purpose of gathering information on the iron oxide nanoparticles’ morphology and size. It was carried out using a scanning electron microscope (Zeiss EVO, Dublin, CA, USA) at 10 kV. As part of this comparison, the samples were photographed at different magnifications and evaluated using the freeware Image-J v1.53q (NIH, Bethesda, MD, USA) [42].

#### 2.3.4. Transmission Electron Microscopy (TEM)

A TEM study was conducted to examine the crystalline characteristics of the size of nanoparticles. The observations were made at 200 kV through a TEM (Talos S200 microscope, FEI, Hillsboro, OR, USA). Images of GS-NPs and CS-NPs were evaluated through Image-J 1.53q free software [42].

### 2.4. Functional Properties

#### 2.4.1. Antioxidant Activity

Iron oxide nanoparticles in this study were tested for their antioxidant activity using the DPPH method (free radical DPPH). This method involves the scavenging of free radicals through DPPH. Thus, the DPPH test can determine the number of free radicals trapped by an antioxidant. It was conducted following the protocol described in previous work [42]. Briefly, 2 mL of nanoparticle dispersion (4, 2, 1, 0.5, 0.25, 0.125 mg/mL in DMSO) was mix-agitated with 2 mL of DPPH. Then, their absorbances were recorded at 517 nm after incubation for 0.5 h. An equal volume of DPPH solution and DMSO was used as a control solution.

The necessary anti-radical that could cause 50% inhibition of DPPH (*IC_50_*) was determined by measuring the inhibition percentage (*IC*) for each concentration using GraphPad Prism 9 software (GraphPad Prism for Windows version 9.0.0, San Diego, CA, USA).

#### 2.4.2. Antimicrobial Activity

The GS-NPs were tested for their antimicrobial activity on agar plates according to Behera et al. (2012) with a few modifications [45]. The antibacterial properties of GS-NPs were tested against two pathogen types: *Staphylococcus aureus* (*S. aureus* gram+) and *Escherichia coli* (*E. coli* gram−) inoculated on agar gels. To this end, 20 μL of dispersion (50 mg/mL) of each selected GS-NPs (the nanoparticles were dispersed in distilled water by ultrasound for 1/3 h) was dropped in holes of 7 mm diameter, which were made using a cone-shaped pipette as a perforator. A standard was prepared by filling wells of 2 mm in diameter with 10 µL of Gentamicin antibiotic (50 µg/mL).

Diameters of inhibition zones surrounding the GS-NPs solutions were measured after 24–48 h of incubation time at 37 °C by Image-J 1.53q (Free software) to assess the antibacterial activity of the different samples [43].

### 2.5. Statistical Analysis

At least three replicates were performed to achieve a reliable comparison. Standard deviations and significant differences were calculated for the statistical analysis using GraphPad Prism9 and IBM SPSS Statistics (version 26.0.0 software). One-way ANOVA was used to signify the differences between observations. These significances were obtained through HSD Tukey statistical analysis, with *p* < 0.05 (95% confidence interval).

## 3. Results and Discussion

### 3.1. Nanoparticle Characterization

#### 3.1.1. XRD

The X-ray diffractograms are shown in Figure 1 (2θ and planes in red indicate the magnetite phase and 2θ and planes in black indicate the hematite phase). Further investigations were conducted on the effects of calcination time and temperature on crystallinity, crystallite size, and phase composition. Regarding the green synthesis (Figure 1a), the diffraction peaks of the GS-NPs calcined at 200 °C for 2 h could be attributed to 98.3% of polycrystalline structures of magnetite (36.2% monoclinic, 32.6% trigonal with hexagonal axis, and 29.3% cubic structures) and 1.7% of cubic hematite (Table 1 and Table 2). These proportions were extracted from (JCPDS nº. 00-153-2800, 00-152-8611, 00-900-7706, 00-900-2673, and 00-210-1535 standard iron oxide powder diffraction pattern) [46,47,48,49,50].

At 4 h of calcination, the diffraction peaks were attributed to 95.3% of polycrystalline magnetite (9.2% monoclinic, 50.0% trigonal with hexagonal axis, and 36.1% cubic structure) and 4.7% of cubic hematite (Table 1 and Table 2) (JCPDS nº. 00-153-2800, 00-152-8611, 00-152-6955 and 00-900-9768 standard iron oxide powder diffraction pattern) [46,50,51,52]. A higher increase in calcination time (5 h) resulted in the formation of 100% magnetite with crystal systems (44.9% monoclinic and 55.1% trigonal with hexagonal axis) (Table 1 and Table 2) (JCPDS nº. 00-153-2800, and 00-152-6955 standard iron oxide powder diffraction pattern) [46,51].

On the other hand, the increase in calcination temperature caused a pronounced change in the Fe_x_O_y_-NP type, which resulted in a greater proportion of hematite. For example, the diffraction peaks of GS-NPs calcined at 300 °C for 2, 4, and 5 h were attributed to 55.0, 57.1, and 86.6% of polycrystalline hematite, respectively (Table 2) [46,47,50,51,53,54], while the diffraction peaks of the GS-NPs calcined at 500 °C for 2, 4 and 5 h were assigned to 25.4, 91.3, and 62.4% of monoclinic, cubic, and trigonal with hexagonal axis hematite [46,47,49,50,51,52,55,56,57,58,59,60].

The chemically synthesized nanoparticles were mostly hematite (Table 1 and Table 2). The diffraction peaks of the CS-NPs calcined at 200 ° C for 2, 4, and 5 h were attributed to 91.1, 92.9, and 95.5% of the polycrystalline structure of hematite, respectively (Figure 1b and Table 2) [46,56,61,62,63]. A temperature increment to 300 °C generated a change in nanoparticles, finding 88.6, 93.5, and 98.1% of the polycrystalline structure of hematite for the calcination times 2, 4, and 5 h, respectively (Figure 1b and Table 2) [46,51,56,61,64]. The further calcination at higher temperatures (500 °C) resulted in nearly pure hematite. Thus, the diffraction peaks were attributed to 96.9, 100, and 98.7% of the polycrystalline structure of hematite for the calcination times 2, 4, and 5 h, respectively (Figure 1b and Table 2) [46,51,56,62,64].

These results are in good accordance with those that have been documented in the literature. Among them, Jafari et al. noted a transition phase of iron oxide nanoparticles from Fe_3_O_4_ to Fe_2_O_3_ when the particles were annealed in the range of 250–550 °C [65]. Shangguan et al. observed an increase in Fe_2_O_3_ proportion by increasing the calcination time (0.5–2.5 h) and calcination temperature in the range of 450–500 °C [66]. Rusul et al. also obtained polycrystalline α-Fe_2_O_3_ nanoparticles at elevated annealing temperatures (500 and 700 °C) [67]. In this way, it seems that the mechanism for the production of iron oxide nanoparticles is due to the reduction of iron ions by phenolic compounds (green) or OH (chemical). Thereafter, two oxidation-reduction reactions were suggested: 1-Reduction and precipitation of iron ions Fe^+3^ of iron chloride (FeCl_3_·6H_2_O) in the presence of phenolic compound (-OH), where the reduction of iron ions to magnetite will normally take place at low calcination temperature and time, according to Equation (1) [65]:3Fe^+3^_(aq)_ + 8OH^−^ → Fe_3_O_4 ↓_ + 4H_2_O(1)

Additionally, and/or the reduction of Fe_2_O_3_ nanoparticles with hydrogen to precipitate Fe_3_O_4_ nanoparticles, according to Equation (2) [68]:3Fe_2_O_3_ + H_2_ ⇌ 2Fe_3_O_4 ↓_ + H_2_O(2)

2-Oxidation of iron germs growing in an aqueous solution (resulted in Fe_2_O_3_) through replacement and armoring process with the increase in calcination temperature and time in the presence of oxygen atoms, according to Equation (3) [65]:4Fe_3_O_4_ + O_2_ → 6Fe_2_O_3_(3)

In this way, magnetite fractions, Fe_3_O_4_-NPs, were more abundant in the GS-NPs, due to the great capacity of phenolic compounds to reduce more Fe^+3^ ions to Fe^+2^ (Fe_3_O_4_-NPs). The absence of phenolic compounds (CS-NPs) generated the oxidation of Fe^+2^ (Fe_3_O_4_) ions into Fe^+3^ (Fe_2_O_3_) [69]. Several studies have reported similar results [70,71].

Regarding the size and crystallinity of GS-NPs and CS-NPs, a summary of these parameters, as well as the proportions of Fe_x_O_y_-NPs, is presented in Table 1. It was observed that the increase in calcination temperature and time increased the size of the nanoparticles (case GS-NPs). The crystallinity degree was higher when the nanoparticles’ size was smaller (Table 1) [72,73]. The decrease in nanoparticle size at low calcination temperature was attributed to oxygen vacancies created on the surface. On the other hand, the increase in calcination temperature up to 300 °C led to an increase in the particle size due to the Ostwald Ripening mechanism (i.e., the particles with larger sizes grow at the expense of the smaller ones) [74,75]. The formation of relatively smaller nanoparticles happens during the calcination rather than the loss of water in the particles [76].

A decrease in size was also reported by Mornani et al., who investigated the effect of calcination temperatures on the magnetic iron oxide nanoparticles using a chemical method in the range of 400–650 °C [18]. An increase in nanoparticle sizes was also observed by Koupaei et al. when the calcination time increased [77,78,79].

Nevertheless, the decrease in size after increasing the calcination time to 5 h at 500 °C (GS-NPs) and at 200, 300, and 500 °C (CS-NPs) (Table 1) is probably due to the oxidation of magnetite at the grain surface to form smaller hematite (Fe_2_O_3_) [79,80].

#### 3.1.2. FTIR

Figure 2a,b shows the general FTIR spectra of GS-NPs and CS-NPs, respectively. The bonds between 3500 and 3420 cm^−1^ are a result of the vibrational stretching of the OH-phenolic compound present in the extract (GS-NPs) and OH of NaOH (CS-NPs) [81].

The sharp peaks at 2912–2838 cm^−1^ are associated with the C-H extension. The band around 1630 cm^−1^ indicates C=C stretching vibration or aromatic ring deformation. Around 1733 cm^−1^, the bands could be related to the C=O of ketones, aldehydes, and esters. Ester groups show also one band around 1380 cm^−1^ [69,82]. The asymmetric stretching vibrations of C-O-C and C-O, corresponding to polyphenolic compounds, could be observed between 1038 and 1071 cm^−1^ and between 1201 and 1248 cm^−1^, respectively [69]. Furthermore, the bonds between 1159 and 1104 cm^−1^ belong to C–O–H in polyphenolic compounds [69]. The reduction of the precursor (FeCl_3_·6H_2_O) by phenolic groups (–OH) results in the segmentation of the band around 1642 cm^−1^ into three peaks around 1652, 1632, and 1622 cm^−1^. Figure 3a,b shows especially the fingerprint of FTIR spectra of GS-NPs and CS-NPs, respectively.

Magnetic Fe-O bands could be observed in the range of 800–400 cm^−1^, where they can be interpreted as magnetite (Fe_3_O_4_) and γ-Fe_2_O_3_/α-Fe_2_O_3_. In this range, compared with maghemite, magnetite (Fe_3_O_4_) bands could be localized between 640 and 570 cm^−1^, obtaining shoulders around 698 cm^−1^ and 447 cm^−1^, corresponding to octahedral and tetrahedral positions of the Fe-O bond, respectively [15,41,83,84,85]. As for the bands at 590–570 cm^−1^, they may result from α-Fe_2_O_3_ reduction [86]. Nevertheless, Fe_3_O_4_ FTIR spectra also revealed absorption bands at 1627 cm^−1^ and 1390 cm^−1^, as well as peaks around 1286 and 1084 cm^−1^ [87]. It is noteworthy that the Fe^3+^ (Fe_2_O_3_) reduction to Fe^2+^ (Fe_3_O_4_) is due to the presence of phenolic compounds in the extracts, where these bands were strongly observed in GS-NPs FTIR spectra (Figure 3a) [69]. However, peculiar bonds around 740–640 cm^−1^ may correspond to the maghemite phase (γ-Fe_2_O_3_) formed by oxidizing magnetite during its synthesis [85,88,89]. On the other hand, bands around 562, 540, and 462 cm^−1^ may indicate the hematite phase (α-Fe_2_O_3_) [89,90]. Moreover, Fe-O vibrations of crystalline α-Fe_2_O_3_ could be observed at 1136 cm^−1^ [70,91]. These bands were clearer in the FTIR spectra of CS-NPs (Figure 2b).

#### 3.1.3. SEM

The SEM images in Figure 4a,b illustrate the morphology and size distribution histograms of the GS-NPs and CS-NPs, respectively.

As can be seen, GS-NPs exhibited different surface morphology between spherical, cubic, rhombohedral, and some hexagonal structures, with homogeneous size distributions. According to visual observation, the GS-NPs calcined at lower temperatures (200–300 °C) and times (2–4 h) tended to be slightly agglomerated or aggregated. This could be due to the fact that remaining phenolic compounds may interact with nanoparticles’ surfaces, or their presence on the particles’ surface could produce H-bonds in bioactive molecules, giving the appearance of aggregates [1,69,92,93]. Meanwhile, the CS-NPs were more agglomerated/aggregated due to the lack of stabilizing agents [21], and the strong interactions between nanoparticles were caused by higher reactivity of the surface [94].

GS-NPs and CS-NPs were measured with respective average diameters from 5.8 to 26.0 ± 1.7 nm and 19.2 ± 0.1 to 37.0 ± 1.0 nm (Table 1). Both these results and the XRD results were very good. Further confirmation of the results was provided by TEM analysis.

#### 3.1.4. TEM

Figure 5 shows TEM images of the selected GS-NPs calcined at 200 and 300 °C for 2 and 4 h with clear morphology. In addition, the size distribution histograms were fitted using a normal curve with their average diameters.

As can be observed, the GS-NPs calcined at 200 and 300 °C for 2 h seemed to be better dispersed and slightly aggregated ultrasmall cubic with some hexagonal structures. In this way, the reducing agent (phenolic compound -OH^−^) remaining on the surface of the nanoparticles produces a substantial increase in iron hydroxide seeds $\mathrm{Fe}{\left(\mathrm{OH}\right)}_{2,3}^{2,3^{-}},$ leading to the formation of Fe_x_O_y_-NPs, which reduces the growth rate and the interparticle reactions [95]. At 4 h of calcination, some nanoparticles seemed to be elongated at 200 °C. On the other hand, the nanoparticles calcined at 300 °C for 4 h showed more agglomeration or aggregation with quasi-spherical, cubic, and hexagonal structures. This may indicate a competitive relationship between the extract’s functional groups (phenolic compounds) and iron ions on the Fe_3_O_4_-NPs surface; thus, the oxidation rate will also increase [96]. These results confirm the results obtained by SEM and XRD.

### 3.2. Functional Properties

#### 3.2.1. Antioxidant Activity

GS-NPs and CS-NPs were examined in terms of their capacity to trap free radicals within the body. The ability to scavenge DPPH was measured by *IC_50_* (µg/mL), where a lower *IC_50_* represents better antioxidant activity. A summary of the *IC_50_* values is provided in Table 1. It can be noticed that the highest antioxidant activity was achieved by the green synthesis (GS-NPs calcined at 200–300 °C for 2–4 h), whereas the most significant antioxidant activity was achieved by those calcined at 300 °C for 2 h with *IC_50_* = 8 ± 2 µg/mL (Table 1). These values were smaller than the *IC_50_* value of gallic acid as a standard, which was found to be 140 ± 60 µg/mL in this work. This higher antioxidant activity of these nanoparticles may be the result of the simultaneous polyphenolic activity (as antioxidants) and catalytic activity of GS-NPs [97]. Phytochemicals and iron ions may serve as antioxidants by releasing oxygen or transferring single electrons and hydrogen atoms [21,98,99]. In this sense, phytochemicals present in the matrix enhance the formation of smaller nanoparticle seeds with higher antioxidant properties [100]. Therefore, small-size nanoparticles may lead to higher antioxidant activity, as it was possible to observe that this system (GS-NPs calcined at 200–300 °C for 2–4 h) had smaller nanoparticle sizes (Table 1, Figure 5) [69]. This could also be due to the higher proportion of magnetite (Fe_3_O_4_) and its crystallinity [97,101]. The increase in the calcination time to 5 h resulted in lower antioxidant activity, which may be due to the same reason found with the increase in calcination temperature to 500 °C. Thus, an increment in the calcination temperature results in phytochemical degradation, which decreases their antioxidant ability [20], as well as the stereoselective nature of the bioactive compound present on the Fe_x_O_y_-NPs surface when they act as antioxidant agents [102]. On the other hand, the CS-NPs showed lower antioxidant activity during the entire study, which reflects the inability of the stabilizing agent and their larger size [21]. Similar antioxidant properties have been reported in the literature [13].

#### 3.2.2. Antimicrobial Activity

Table 3 presents the inhibition area of the tested GS-NPs as a function of their significant parameter (diameter, mm). These results were similar to those obtained with Gentamicin (50 µg/mL). As is shown in Figure 6, all tested GS-NPs demonstrated moderate bacterial inhibition for both pathogenic *S. aureus* gram+ and *E. coli* gram-bacteria.

As a result of their smaller size (D_DRD_ ≈ 8–16 nm, D_SEM_ ≈ 6–19 nm, and D_TEM_ ≈ 5–13 nm), GS-NPs were capable of inhibiting the replication of bacterial DNA effectively. In this way, small zero-valent iron oxide nanoparticles could penetrate and inactivate the bacterial membrane by oxidizing the intercellular matrix, which results in oxidative stress (OS), damaging the bacterial cell membrane [103,104]. Oxidative stress (OS) is generated by many reactive oxygen species (ROS), such as hydroxyl radicals (–OH), superoxide radicals (O^2−^), singlet oxygen (^1^O_2_), and hydrogen peroxide (H_2_O_2_), which has the potential to damage both proteins and DNA. Thus, bacteria are especially vulnerable to ROS [31,105]. However, the high antibacterial properties of GS-NPs may also be related to their high crystallinity (98.6%), which enables the released iron ions (Fe^+2^ and Fe^+3^) to collide with negatively charged bacterial membranes, leading to the destruction of their protein structures [106,107]. GS-NPs inhibited both *S. aureus* and *E. coli*, which could be potential sources of reactive oxygen species. Similar results have also been cited in the literature [108,109,110]. Several other mechanisms may contribute to the antibacterial activity of metal oxide nanoparticles. In addition to the previously mentioned (ROS generation), bacteria could be damaged by the diffusion of isolated nanoparticles in the bacterial cell membrane and the electrostatic interactions between nanoparticles and bacteria surface [111]. Nevertheless, it was noted that the GS-NPs calcined at a lower temperature (GS-NPs _200°C 2h_) exhibited more antibacterial activity than those calcined at a higher calcination temperature (GS-NPs _300°C 2h_). In this way, the GS-NPs calcined at higher calcination temperatures were slightly agglomerated; therefore, these agglomerations do not allow them to diffuse into the cells and cause damage from within.

## 4. Conclusions

In this study, the use of *Phoenix dactylifera L.* (*PDL*) extract allowed for the obtaining of suitable GS-NPs. The results corroborated the importance of the calcination stage (calcination temperature and time). Consequently, low calcination temperatures and short calcination times prevent the degradation of active agents. This leads to the formation of nanoparticles of smaller sizes, fewer polycrystalline structures, and better functional properties. Alternatively, the increase in calcination temperature or time may result in larger grains, heterogeneous crystal phases, and phytochemical degradation due to the stereoselective nature of the bioactive compound present on the nanoparticles’ surface when they act as functional agents. The impact of this process stage allows for determining the final phase, purity, and functionality of the final product. Nevertheless, impurities can be removed through the washing process. Iron oxide nanoparticles developed through green synthesis using *PDL* extract exhibited excellent antioxidant and antibacterial properties, making it a promising alternative as an antibacterial drug that can replace the current antibiotics. It is our knowledge that no study has reported an *IC_50_* value < 100 µg/mL for the green synthesized iron oxide nanoparticles using *Phoenix dactylifera L.,* which may suggest their potential applications without being toxic or hazardous. Therefore, the GS-NPs can potentially substitute chemical nanomaterials due to their numerous advantages, including easy preparation, cost-effectiveness, environmental friendliness, and high suitability for the implementation of metallic nanomaterials in many applications. However, further investigation is needed concerning nanoparticle cytotoxicity and migration in biomedical and food applications. Furthermore, future studies will examine the in vivo magnetic response and stability of these nanoparticles, taking into account their expected environmental friendliness.

## Figures and Tables

**Figure 1 materials-16-01798-f001:**
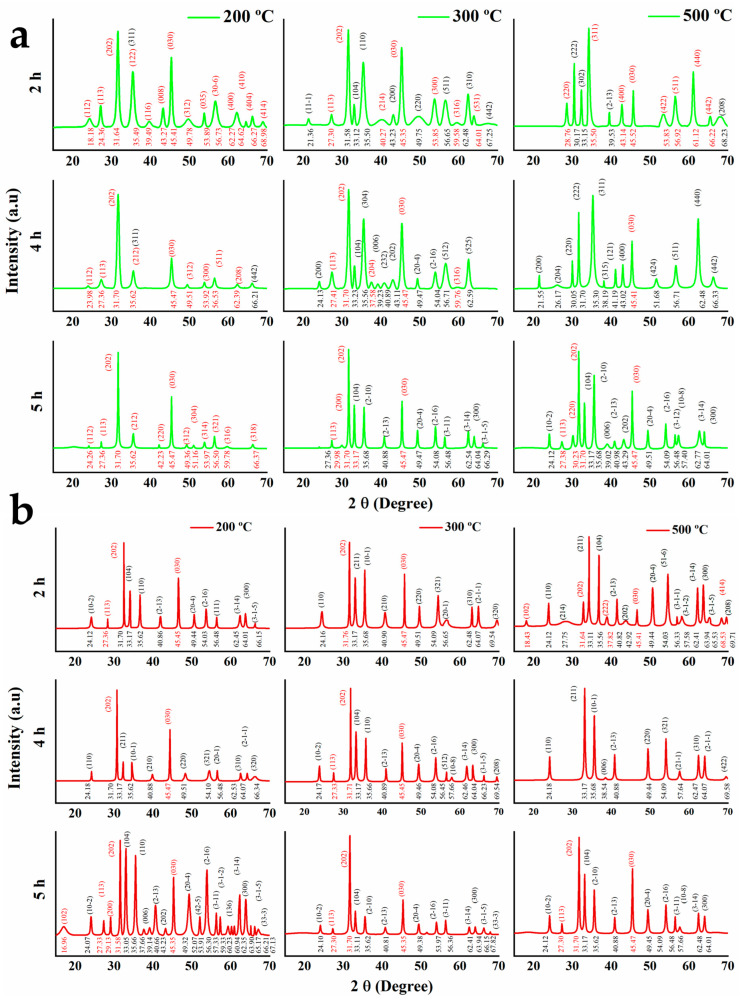
XRD patterns of the nanoparticles (JCPDS standard) calcined at different temperatures (200, 300, and 500 °C) and times (2, 4, and 5 h): (**a**) GS-NPs profile using aqueous extracts of *PDL* and (**b**) CS-NPs profile using NaOH.

**Figure 2 materials-16-01798-f002:**
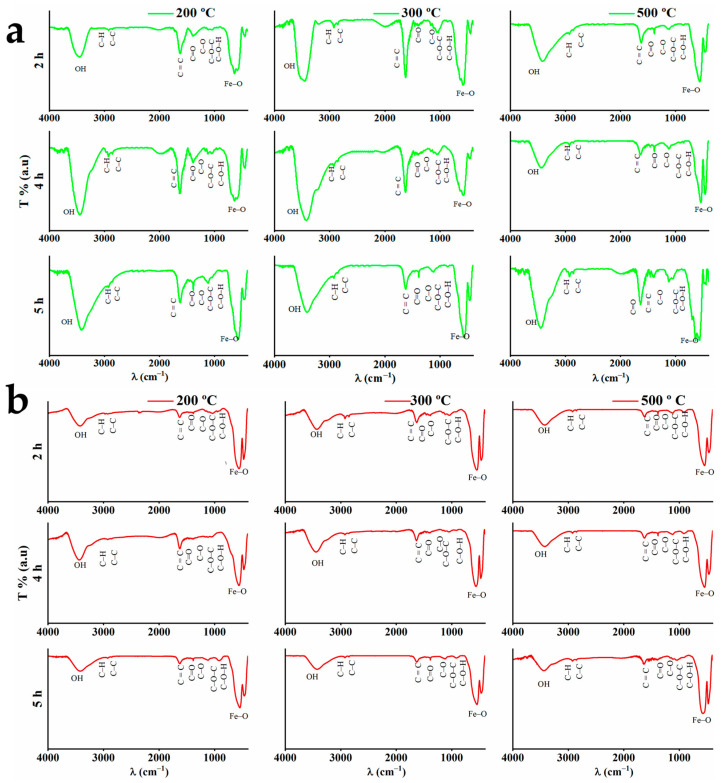
General FTIR spectra (4000–400 cm^−1^) of the nanoparticles calcined at different temperatures (200, 300, and 500 °C) and times (2, 4, and 5 h): (**a**) GS-NPs profile using aqueous extracts of *PDL* and (**b**) CS-NPs profile using NaOH.

**Figure 3 materials-16-01798-f003:**
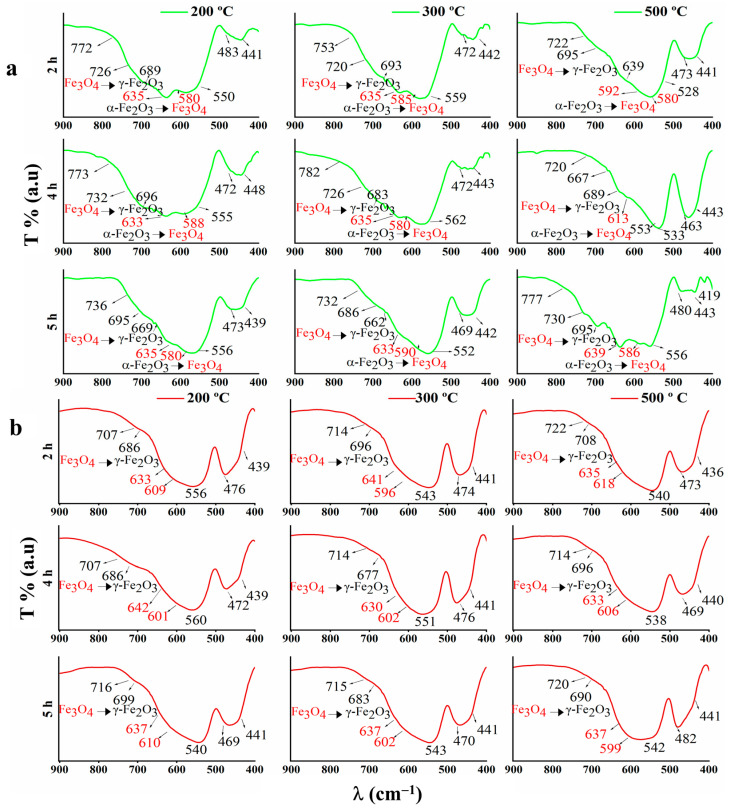
Fingerprint region of FTIR spectra in the (900–400 cm^−1^) of the nanoparticles calcinated at different temperatures (200, 300, and 500 °C) and times (2, 4, and 5 h): (**a**) GS-NPs profile using aqueous extracts of *PDL* and (**b**) CS-NPs profile using NaOH.

**Figure 4 materials-16-01798-f004:**
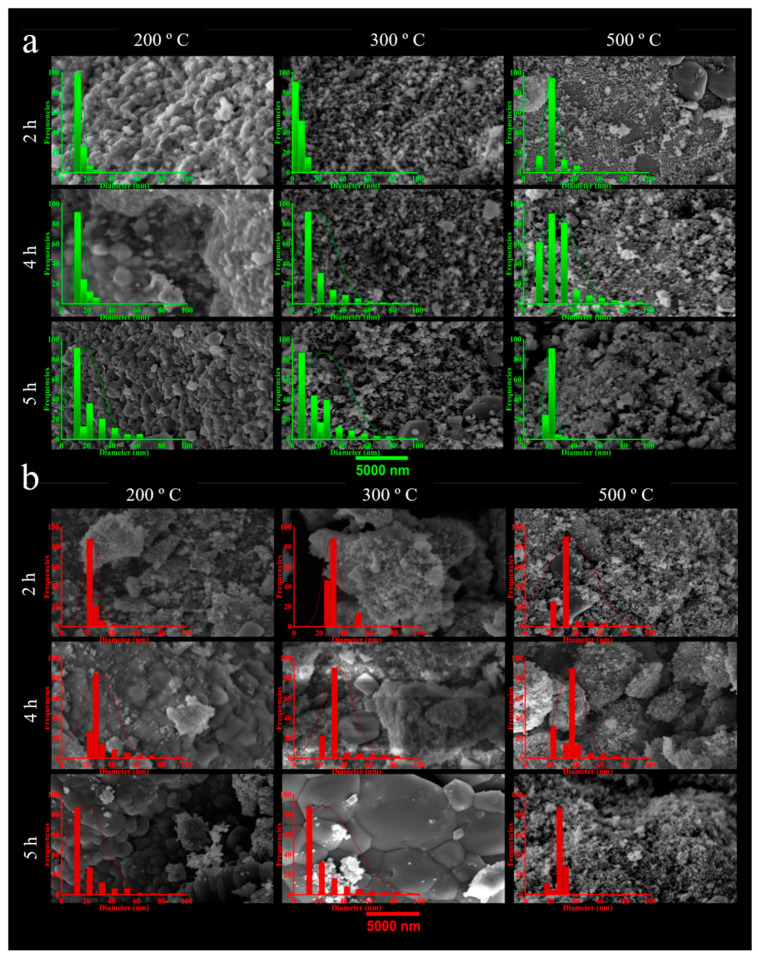
Scanning electron microscopy (SEM) of the nanoparticles calcinated at different temperatures (200, 300, and 500 °C) and times (2, 4, and 5 h): (**a**) GS-NPs profile using aqueous extracts of *PDL* and (**b**) CS-NPs profile using NaOH.

**Figure 5 materials-16-01798-f005:**
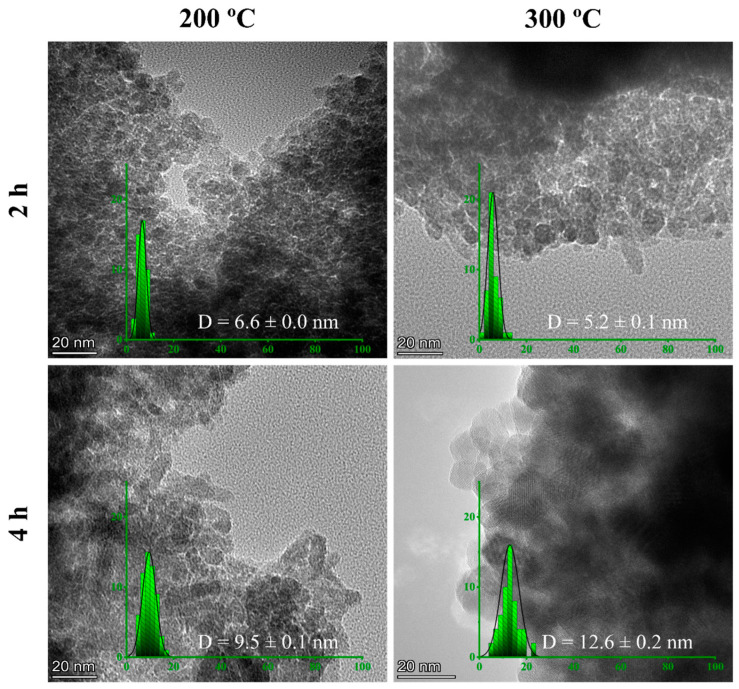
Transmission electron microscopy (TEM) of selected GS-NPs nanoparticles calcinated at temperatures (200 and 300 °C) and times (2 and 4 h).

**Figure 6 materials-16-01798-f006:**
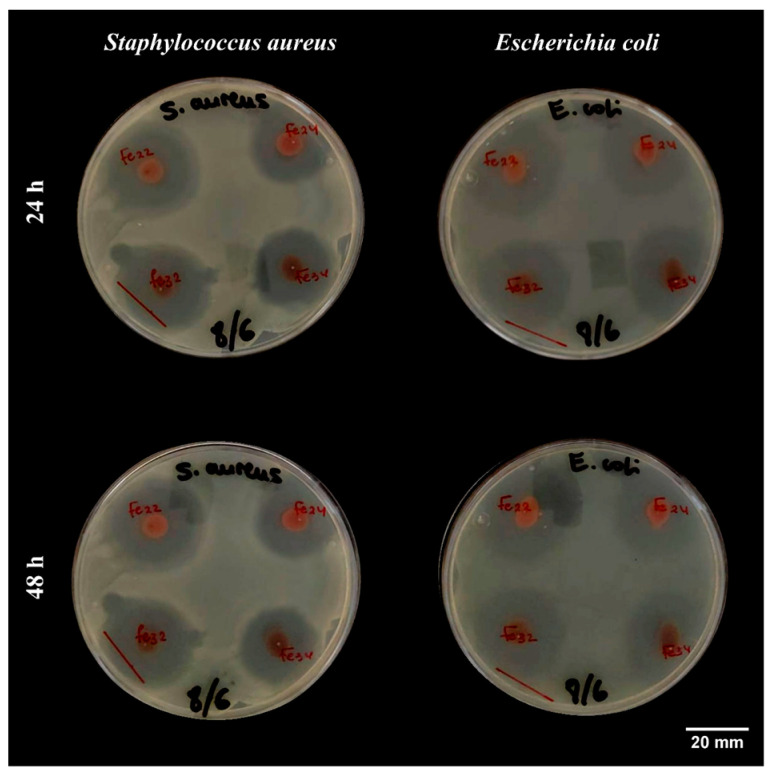
The inhibition zones (diameters) produced by selected GS-NPs Fe_22_. GS-NPs calcinated at 200 °C for 2 h, Fe_24_. GS-NPs calcinated at 200 °C for 4 h, Fe_32_. GS-NPs calcinated at 300 °C for 2 h and Fe_34_. GS-NPs calcinated at 300 °C for 4 h against *Staphylococcus aureus* (*S. aureus*) and *Escherichia coli* (*E. coli*) over 48 h.

**Table 1 materials-16-01798-t001:** Results obtained for different parameters of GS-NPs using *PDL* extract and CS-NPs using the chemical method at different temperatures (200, 300, and 500 °C) and times (2, 4, and 5 h): average size in nm (from XRD measurements and SEM images), crystallinity percentage (Crl, %), composition (proportion of Fe_2_O_3_ and Fe_3_O_4_), and antioxidant activity (DPPH *IC_50_* in µg/mL).

	GS-NPs																	
T °C	200 °C						300 °C						500 °C					
Time	XRD (nm)	SEM (nm)	Crl (%)	Fe_2_O_3_ (%)	Fe_3_O_4_ (%)	*IC_50_* (µg/mL)	XRD (nm)	SEM (nm)	Crl (%)	Fe_2_O_3_ (%)	Fe_3_O_4_ (%)	*IC_50_* (µg/mL)	XRD (nm)	SEM (nm)	Crl (%)	Fe_2_O_3_ (%)	Fe_3_O_4_ (%)	*IC_50_* (µg/mL)
2 h	10.2 ± 0.4 ^f^	11.7 ± 0.2 ^f^	98.5 ^a^	1.7	98.3	86 ± 6 ^e^	7.6 ± 0.5 ^f^	5.8 ± 0.1 ^e^	98.4 ^a^	55.0	45.0	8 ± 2 ^f^	22.0 ± 0.3 ^e^	21.8 ± 0.6 ^c^	96.6 ^b^	25.4	74.6	1488 ± 449 ^f^
4 h	11.8 ± 0.6 ^e^	14.2 ± 0.3 ^e^	96.7 ^b^	4.7	95.3	55 ± 2 ^f^	16.3 ± 0.2 ^e^	19.3 ± 0.2 ^d^	98.6 ^a^	57.1	42.9	39 ± 3 ^e^	24.6 ± 0.8 ^d^	26.0 ± 1.7 ^b^	97.0 ^a^	91.3	8.7	4586 ± 343 ^c^
5 h	18.5 ± 1.2 ^d^	16.8 ± 3.6 ^d^	87.6 ^f^	-	100	403 ± 41 ^d^	27.7 ± 0.3 ^d^	23.5 ± 1.8 ^c^	94.6 ^c^	86.6	13.4	109 ± 3 ^d^	23.5 ± 0.6 ^d^	20.8 ± 0.1 ^d^	91.1 ^d^	62.4	37.6	2100 ± 10 ^e^
	**CS-NPs**																	
2 h	29.8 ± 0.8 ^b^	24.2 ± 0.4 ^b^	93.4 ^c^	91.1	8.9	4213 ± 990 ^c^	32.1 ± 1.1 ^b^	25.3 ± 0.9 ^b^	93.6 ^d^	88.6	11.4	4810 ± 165 ^c^	32.3 ± 0.6 ^a^	36.0 ± 1.0 ^a^	91.7 ^d^	96.9	3.1	97,778 ± 2320 ^a^
4 h	31.9 ± 0.3 ^a^	26.3 ± 0.4 ^a^	90.5 ^e^	92.9	7.7	12,602 ± 779 ^b^	39.0 ± 1.7 ^a^	31.8 ± 0.9 ^a^	95.0 ^b^	93.5	6.5	19280 ± 1249 ^a^	31.3 ± 0.1 ^b^	37.0 ± 1.0 ^a^	94.9 ^c^	100.0	-	3735 ± 1650 ^d^
5 h	22.2 ± 0.7 ^c^	19.2 ± 0.1 ^c^	92.4 ^d^	95.5	4.5	23,442 ± 1362 ^a^	29.5 ± 0.9 ^c^	23.0 ± 1.0 ^c^	91.0 ^e^	98.1	1.9	16,550 ± 1208 ^b^	29.6 ± 0.9 ^c^	27.5 ± 0.5 ^b^	85.6 ^e^	98.7	1.3	7100 ± 400 ^b^

Note: Different superscript letters (a–f) within a row indicate significant differences among mean observations (*p* < 0.05).

**Table 2 materials-16-01798-t002:** Results obtained for different parameters of GS-NPs using *PDL* extract and the CS-NPs using the chemical method at different temperatures (200, 300, and 500 °C) and times (2, 4, and 5 h): Crystal systems attributed to hematite (Fe_2_O_3_) and magnetite (Fe_3_O_4_) proportions; Mo: Monoclinic; Tri (hex): Trigonal (hexagonal axis); Tri (rohm): trigonal (rhombohedral axis); Cu: Cubic; Te: tetrahedral.

	GS-NPs																		
T °C		300 °C						300 °C						500 °C					
Time	Phase	MO	Tri (Hex)	Tri (rohm)	Cu	Te	Total	MO	Tri (Hex)	Tri (rohm)	Cu	Te	Total	MO	Tri (Hex)	Tri (rohm)	Cu	Te	Total
2 h	Fe_2_O_3_				1.7		1.7	0.7	0.6	47.9	5.8		55.0	25.4					25.4
	Fe_3_O_4_	36.2	32.6		29.4		98.3	35.3	6.6		3.1		45.0				74.6		74.6
4 h	Fe_2_O_3_				4.7		4.7				7.2	49.9	57.1				91.3		91.3
	Fe_3_O_4_	9.2	50.0		36.1		95.3				42.9		42.9	1.8	1.4		5.5		8.7
5 h	Fe_2_O_3_							0.5	86.1				86.6		62.4				62.4
	Fe_3_O_4_	44.9	55.1				100	7.9	2.5		3.0		13.4	19.0	1.0		17	0.6	37.6
	**CS-NPs**																		
2 h	Fe_2_O_3_	1.5	89.6				91.1		88.6				88.6	3.4	46.7	46.8			96.9
	Fe_3_O_4_	6.3	2.6				8.9	11.4					11.4	2.1	1.0				3.1
4 h	Fe_2_O_3_	2.9	2.5	87.5			92.9	0.7	92.8				93.5	2.2	2.8	95.0			100
	Fe_3_O_4_	2.9	4.2				7.1	5.7	0.8				6.5						
5 h	Fe_2_O_3_	6.1	89.4				95.5	1.5	96.6				98.1	60.3	36.1	2.3			98.7
	Fe_3_O_4_	4.5					4.5	1.1	0.8				1.9	1.3					1.3

**Table 3 materials-16-01798-t003:** The inhibition diameters produced by selected GS-NPs nanoparticles calcinated at temperatures (200 and 300 °C) and times (2 and 4 h) against *Staphylococcus aureus* (*S. aureus*) and *Escherichia coli* (*E. coli*) over 48 h.

Test Time (h)	Sample	*S. aureus*	*E. coli*
24	Gentamicin	28.3 ± 0.3 ^a^	30.4 ± 0.7 ^A^
	GS-NPs _200°C 2h_	23.3 ± 0.7 ^b^	24.9 ± 0.4 ^B^
	GS-NPs _200°C 4h_	19.2 ± 0.2 ^d^	20.8 ± 0.1 ^D^
	GS-NPs _300°C 2h_	21.8 ± 0.7 ^c^	24.2 ± 0.3 ^BC^
	GS-NPs _300°C 4h_	18.4 ± 0.9 ^d^	23.5 ± 1.0 ^C^
48	Gentamicin	26.6 ± 0.1 ^a^	28.1 ± 0.4 ^A^
	GS-NPs _200°C 2h_	23.6 ± 0.6 ^b^	25.1 ± 0.3 ^B^
	GS-NPs _200°C 4h_	17.1 ± 0.2 ^d^	20.2 ± 0.2 ^D^
	GS-NPs _300°C 2h_	20.8 ± 0.3 ^c^	21.4 ± 0.0 ^C^
	GS-NPs _300°C 4h_	15.7 ± 0.7 ^e^	21.2 ± 0.5 ^C^

Note: Different superscript letters (a–e and A–D) within a column in a time raw illustrate a significant difference between mean observations (*p* < 0.05).

## Data Availability

The data presented in this study are available upon request from the corresponding author.

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
