# Peer review of "Effect of Calcination Temperature and Time on the Synthesis of Iron Oxide Nanoparticles: Green vs. Chemical Method"

_materials, 2023, doi:10.3390/ma16051798_

Round 1

Reviewer 1 Report

materials-2189963-Effect of calcination temperature and time on the synthesis of 2 iron oxide nanoparticles: Green vs. chemical method

Abstract, line 10-14: this is very general information, should be moved to introduction section.

Introduction: line 91-101, authors are advised to provide information relating to the objective of the work, the same information could be move to Materials and Methodology.

2. Materials and Methodology, line 107, and used in this research……. better to write in the beginning of the sentence for instance, in this study, following chemical/materials……………. were used.

3. Results, should be results and discussion.

Conclusions should be concise, repeated information should be evaded.

-English language should be improved.

Author Response

Reviewer: 1

Comments:

materials-2189963-Effect of calcination temperature and time on the synthesis of 2 iron oxide nanoparticles: Green vs. chemical method

The authors gratefully acknowledge the reviewer’s comments and suggestions. All of them have been taken into consideration to improve the quality of the manuscript.

  1. Abstract, line 10-14: this is very general information, should be moved to introduction section. Introduction: line 91-101, authors are advised to provide information relating to the objective of the work, the same information could be move to Materials and Methodology.

We gratefully appreciate this observation. Accordingly, we have considered re-writing this observation with a short consistent introduction (Lines 10-17). “Nowadays, antioxidants and antibacterial activity play an increasingly vital role in biosystems due to the biochemical and biological reactions that involve free radicals and pathogen growth, which occur in many systems. For this purpose, continuous efforts are being made to minimize these reactions, including the use of nanomaterials as antioxidants and bactericidal agents. Despite such advances, iron oxide nanoparticles still lack knowledge regarding their antioxidant and bactericidal capacities. This includes the investigation of biochemical reactions and their effects on nanoparticle functionality. In green synthesis, active phytochemicals give nanoparticles their maximum functional capacity and should not be destroyed during synthesis.” The information relating to the objective of the work has been provided according to the reviewer’s comment (Lines 91-104).

  1. Materials and Methodology, line 107, and used in this research……. better to write in the beginning of the sentence for instance, in this study, following chemical/materials……………. were used.

We appreciate the reviewer for this valuable suggestion. This section has been changed according to the reviewer's suggestion (Lines 107-111).

  1. Results, should be results and discussion.

We thank the reviewer for this comment. We have considered this suggestion and we included it (Line 187).

  1. Conclusions should be concise, repeated information should be evaded.

We would like to thank the reviewer again for his/her comments. Considering this comment, repeated information has been avoided and the conclusion has been improved.

  1. English language should be improved.

The authors apologize for any grammatical or other errors in the manuscript. Accordingly, thorough revisions and corrections have been made to the manuscript. 

Reviewer 2 Report

The topic addressed is current, it opens new ways in obtaining nanomaterials that can be used in current therapeutic practice. I recommend the authors to perform in vivo testing of the various formulations. For the current article, I recommend revaluing the conclusions section, by extracting some current conclusions, because in the current form it is presented more like a discussion section.

Author Response

Reviewer: 2

Comments:

The topic addressed is current, it opens new ways in obtaining nanomaterials that can be used in current therapeutic practice. I recommend the authors to perform in vivo testing of the various formulations. For the current article, I recommend revaluing the conclusions section, by extracting some current conclusions, because in the current form it is presented more like a discussion section.

We thank the reviewer for doing an in-depth analysis of our work and providing such useful suggestions. We agree that the performance of in-vivo testing would certainly be interesting. Therefore, the reviewers' suggestions have been incorporated into the conclusion section to be considered in future studies. Furthermore, the conclusion section has been improved regarding the reviewers' comments (Lines 432-434) “It is our knowledge no study has reported an IC50 value < 100 µg/mL for the green synthesized iron oxide nanoparticles using Phoenix dactylifera L., which may suggest their potential applications without being toxic or hazardous.” and (Lines 438-4341): “However, further investigation is needed concerning nanoparticle cytotoxicity and migration in biomedical and food applications. Furthermore, future studies will examine the in-vivo magnetic response and stability of these nanoparticles, taking into account their expected environmental friendliness”.

.

Reviewer 3 Report

Comments and suggestions are attached.

Author Response

Reviewer: 3

Comments:

This manuscript deals with the effect of calcination temperature and time on synthesis of iron oxide nanoparticles by a green and a chemical method. In this work the authors highlight the importance of green synthesis of iron oxide nanoparticles due to their excellent antioxidant and antimicrobial activities. The manuscript contains very valuable information of interest for the wide scientific community interested in this kind of compounds. In my opinion the manuscript could be considered for publication in present form with only some small observations:

The authors gratefully acknowledge the reviewer’s comments and suggestions. All of them have been taken into consideration to improve the quality of the manuscript.

  1. Table 2: How was calculated the proportions of crystals systems (Mo, Tri (hex), Tri (rohm), Cu, Te) for both hematite (Fe2O3) and magnetite (Fe3O4).

We gratefully appreciate the query. All proportions calculated in this study were obtained from JCPDS standard iron oxide powder diffraction pattern as referenced in the manuscript. This has been mentioned in the manuscript (Lines 197-198): “These proportions were extracted from (JCPDS nº. 00-153-2800, 00-152-8611, 00-900-7706, 00-900-2673 and 00-210-1535 standard iron oxide powder diffraction pattern) [46–50].”

  1. Some phrases are too long for instance: lines 307-312. In summary, I will recommend this paper for publication after corrections.

We appreciate the reviewer for this observation, and we gratefully appreciate the recommendation. According to this observation, long phrases have been shortened (Lines 264-267) (Lines 289-291) (Lines 302-312) (Lines 308-310).

Reviewer 4 Report

In the present research article, Johar et al. have discussed the effect of calcination temperature and time on the synthesis of iron oxide nanoparticles via the Green vs. chemical method. The authors have explained the research problem with detailed discussion and reasonable conclusion. However, some points still need to be clarified. Thus, I recommend a minor revision before considering the possible publication of this manuscript.

1.    The novelty of the work should be highlighted.

2.    Why are the TEM micrographs for CS-NPs and their discussion not included?

3.   Microbiological methods need to be described. There is no description of the minimum inhibition concentration (MIC).

4.   It is suggested to include the results for the antimicrobial activity of the plant extract as well as CS-NPs.

5. The authors have mentioned the reactive oxygen species, damaging both proteins and DNA of the bacterial cells. However, there is no direct evidence about it.

Author Response

Reviewer: 4

Comments:

In the present research article, Johar et al. have discussed the effect of calcination temperature and time on the synthesis of iron oxide nanoparticles via the Green vs. chemical method. The authors have explained the research problem with detailed discussion and reasonable conclusion. However, some points still need to be clarified. Thus, I recommend a minor revision before considering the possible publication of this manuscript.

The authors gratefully acknowledge the reviewer’s comments and suggestions. All of them have been taken into consideration to improve the quality of the manuscript.

  1. The novelty of the work should be highlighted.

This is a very valuable comment from the reviewer, and we are grateful for it. Accordingly, the following paragraph has highlighted the main novelty of this research (Lines 20-31). ”Nevertheless, no study has extensively examined the effect of photochemical degradation on the efficiency and functionality of nanoparticles, especially those prepared from Phoenix dactylifera L. extract. The active photochemical responsible for donating the nanoparticle its maximum functional capacity should not be degraded or destroyed during the synthesis. Therefore, the main objective of this work was to provide a novel comparison between the green and chemical synthesis of iron oxide nanoparticles and to demonstrate the impact of the calcination stage on their properties and functionality. In this way, Phoenix dactylifera L. (PDL) was used for its polyphenol content, as reported in previous works [39,40]. Factors like calcination temperatures and times, which may influence iron oxide nanoparticles' properties, were investigated. Thus, iron oxide nanoparticles were synthesized by colloidal precipitation using two reducing agents: polyphenols extracted from PDL (green method, GS-NPs) and sodium hydroxide (chemical method, CS-NPs). Nanoparticles of both approaches were investigated under different calcination temperatures (200, 300, 500 °C) and times (2, 3, 5 h). and further characterized by XRD, FTIR, SEM, and TEM. Furthermore, the obtained iron oxide nanoparticles were tested for their antioxidant activity towards DPPH and their antibacterial activity towards Staphylococcus aureus (S. aureus) and Escherichia coli (E. coli).”

  1. Why are the TEM micrographs for CS-NPs and their discussion not included?.

We appreciate the reviewer's query. Using previous techniques for characterizing nanoparticles, we have decided to do this analysis only for optimized samples, thereby shortening the manuscript.

  1. Microbiological methods need to be described. There is no description of the minimum inhibition concentration (MIC).

We thank the reviewer for this valuable comment. We agree that the calculation of the minimum inhibition concentration would certainly be informative. Unfortunately, we used a protocol that only allowed one concentration. Thus, the calculation of MIC needs a series of concentrations to be calculated. We believe that the IC50 calculation would benefit this capacity. We would like to do this calculation in future studies since it is of crucial importance.

  1. It is suggested to include the results for the antimicrobial activity of the plant extract as well as CS-NPs.

We thank the reviewer for doing an in-depth analysis of our study and providing such valuable suggestions. We agree that including the results for the antimicrobial activity of the plant and CS-NPs would certainly be of interest. Unfortunately, at the time the experiments were performed, we did not conduct these experiments and we are unable to do this at the current time. Indeed, we report the optimal conditions with the corresponding characterization to obtain excellent antioxidant and antimicrobial agents. With all respect to the reviewer, we have not felt, at that time, and much to our regret, the need to include this comment. Nevertheless, to meet the reviewer’s concern, some additional detail concerning potential results and the investigation may be carried out in future studies. (Line 430-434):” It is our knowledge that no study has reported an IC50 value < 100 µg/mL for the green synthesized iron oxide nanoparticles using Phoenix dactylifera L., which may suggest their potential applications without being toxic or hazardous” (Line 437-440): “However, further investigation is needed concerning nanoparticle cytotoxicity and migration in biomedical and food applications. Furthermore, future studies will examine the in-vivo magnetic response and stability of these nanoparticles, taking into account their expected environmental friendliness”.

  1. The authors have mentioned the reactive oxygen species, damaging both proteins and DNA of the bacterial cells. However, there is no direct evidence about it.

We would appreciate the query, and we agree with this comment. However, oxidative stress (OS) is one of the mechanisms suggested to inhibit DNA replication. We have mentioned other suggested mechanisms in this section, including iron ions release (Fe+2 and Fe+3) and isolated nanoparticle diffusion. The contribution of OS may not be evident in the case of nanoparticle agglomerations.
